# Effects of Anti-Fibrotic Drugs on Transcriptome of Peripheral Blood Mononuclear Cells in Idiopathic Pulmonary Fibrosis

**DOI:** 10.3390/ijms25073750

**Published:** 2024-03-28

**Authors:** Daisuke Ishii, Takeshi Kawasaki, Hironori Sato, Koichiro Tatsumi, Takuro Imamoto, Keiichiro Yoshioka, Mitsuhiro Abe, Yoshinori Hasegawa, Osamu Ohara, Takuji Suzuki

**Affiliations:** 1Department of Respirology, Graduate School of Medicine, Chiba University, Chiba 260-8670, Japan; 2Department of Pediatrics, Graduate School of Medicine, Chiba University, Chiba 260-8670, Japan; 3Department of Applied Genomics, Kazusa DNA Research Institute, Chiba 292-0818, Japan; 4Synergy Institute for Futuristic Mucosal Vaccine Research and Development, Chiba University, Chiba 260-8670, Japan

**Keywords:** idiopathic pulmonary fibrosis, peripheral blood mononuclear cells (PBMCs), RNA sequencing, transcriptome, pirfenidone, nintedanib

## Abstract

Two anti-fibrotic drugs, pirfenidone (PFD) and nintedanib (NTD), are currently used to treat idiopathic pulmonary fibrosis (IPF). Peripheral blood mononuclear cells (PBMCs) are immunocompetent cells that could orchestrate cell–cell interactions associated with IPF pathogenesis. We employed RNA sequencing to examine the transcriptome signature in the bulk PBMCs of patients with IPF and the effects of anti-fibrotic drugs on these signatures. Differentially expressed genes (DEGs) between “patients with IPF and healthy controls” and “before and after anti-fibrotic treatment” were analyzed. Enrichment analysis suggested that fatty acid elongation interferes with TGF-β/Smad signaling and the production of oxidative stress since treatment with NTD upregulates the fatty acid elongation enzymes *ELOVL6*. Treatment with PFD downregulates *COL1A1*, which produces wound-healing collagens because activated monocyte-derived macrophages participate in the production of collagen, type I, and alpha 1 during tissue damage. Plasminogen activator inhibitor-1 (PAI-1) regulates wound healing by inhibiting plasmin-mediated matrix metalloproteinase activation, and the inhibition of PAI-1 activity attenuates lung fibrosis. DEG analysis suggested that both the PFD and NTD upregulate *SERPINE1*, which regulates PAI-1 activity. This study embraces a novel approach by using RNA sequencing to examine PBMCs in IPF, potentially revealing systemic biomarkers or pathways that could be targeted for therapy.

## 1. Introduction

Idiopathic pulmonary fibrosis (IPF) is a specific form of chronic progressive fibrosing interstitial pneumonia of unknown etiology with poor prognosis. Although not completely understood, the pathobiology of IPF may involve repetitive epithelial injury, subsequent aberrant repair of the injured alveoli, and extensive deposition of the extracellular matrix (ECM) by activated fibroblasts (myofibroblasts) [1,2,3]. Two anti-fibrotic drugs, pirfenidone (PFD) and nintedanib (NTD), are currently used to delay the progression of pulmonary fibrosis [1]. PFD decreases the expression of profibrotic factors such as transforming growth factor-β1 (TGF-β1) and inflammatory cytokines, such as tumor necrosis factor-α, interleukin (IL)-1β, and IL-13 [4,5]. NTD is a tyrosine kinase inhibitor that targets vascular endothelial growth factor (VEGF) receptors 1, 2, and 3; platelet-derived growth factor (PDGF) receptors α and β; and fibroblast growth factor receptors (FGF) 1, 2, and 3 [6]. However, the detailed mechanisms of action of these drugs in progressive fibrosis remain unclear.

Peripheral blood mononuclear cells (PBMCs) are immunocompetent cells classified into monocytes and lymphocytes, though a small percentage of granulocytes are possibly mixed. A recent multicenter cohort study reported that a high monocyte count is a biomarker of poor prognosis in fibrotic diseases, including IPF [7]. Impaired lymphocyte function is associated with respiratory dysfunction in IPF [8]. These studies suggest a close association between peripheral PBMC function and IPF. A recent and growing body of literature has described the key role of immune cells in orchestrating cell–cell interactions, including the crosstalk between epithelial cells and fibroblasts [9,10,11]. However, the relationship between immunocompetent PBMCs and IPF pathobiology remains unclear.

Omics analysis is a useful tool encompassing genomics, epigenomics, transcriptomics, proteomics, and metabolomics and has been widely used to understand diverse polygenic and phenotypic diseases. Two main methods have been used for transcriptome analysis: microarray and RNA sequencing (RNA-seq). A previous study of peripheral PBMCs from patients with IPF using microarray analysis reported 52 genes related to transplant-free survival [12]. Using multiple publicly available microarray datasets of IPF and healthy lung tissues, enrichment analyses of the Kyoto Encyclopedia of Genes and Genomes (KEGG) pathway revealed that differentially expressed genes (DEGs) were enriched in protein digestion and absorption, ECM-receptor interactions, focal adhesion, the PI3K-Akt signaling pathway, amoebiasis, and platelet activation [13]. However, the transcriptome signature of PBMCs in patients with IPF and the potential effects of anti-fibrotic drugs on the transcriptome signature of PBMCs in patients with IPF remain unclear.

This study explored the potential PBMC signature in patients with IPF and the effects of anti-fibrotic drugs on the PBMC signature in patients with IPF using RNA-seq analysis.

## 2. Results

### 2.1. Characteristics of Participants and Changes in FVC 3 Months after PFD and NTD

To explore the potential of the PBMC signature in patients with IPF and the effects of anti-fibrotic drugs on the PBMC signatures in these patients, two groups of six patients with IPF and six healthy controls (HCs) were established. The participants’ characteristics are presented in Table 1. All participants were male, with no significant differences in age between the IPF and HC groups (*p* = 0.69). All patients with IPF and half of the HCs had a history of smoking, none were current smokers, and none had received anti-inflammatory treatment with steroids or immunosuppressants. Other oral medications prescribed to the IPF patients are listed in Appendix A, and the possibility that these medications have any influence on PBMC signatures could not be denied.

Among the six patients with IPF, three were treated with PFD and the other three with NTD. Three months after PFD or NTD administration, the PBMC signature was explored again, and, at the same time, follow-up lung function tests and high-resolution computed tomography (CT) examinations were performed to confirm the inhibitory effects of PFD or NTD on fibrosis progression. The dose of PFD was 1800 mg/day for patients in sample no. 1 and 1200 mg/day for patients in sample Nos. 2 and 3. No side effects were observed in patient sample No. 1, anorexia was detected in patient No. 2, and photosensitivity was detected in patient No. 3. The NTD dose was 300 mg/day for patients in samples 4, 5, and 6. The side effects were diarrhea in treated patients in sample nos. 1 and 2 and none in the treated patient in sample no. 3. Regarding the high-resolution CT findings, in terms of categorizing CT imaging in the evaluation of IPF, the usual interstitial pneumonia (UIP) pattern was observed in the patients of sample Nos. 1, 3, and 4; the probable UIP pattern was observed in those of Nos. 2 and 5; and indeterminate UIP was observed in the patient of sample No. 6. No apparent changes in forced vital capacity (FVC) or high-resolution CT findings were observed three months after the administration of either anti-fibrotic drug.

### 2.2. Comparison of Transcriptome Signatures in PBMCs between IPF and HCs

First, we explored the potential signatures of PBMCs in patients with IPF. RNA-seq libraries of mRNA isolated from PBMCs were prepared from patients with IPF who did not receive anti-fibrotic treatment (n = 6) or HCs (n = 6). The RNA integrity values of all the samples were >9, and 26,472 mRNA genes were identified. After quality control, 12,347 genes were analyzed.

Principal component analysis (PCA) comparing gene expression levels between patients with IPF and HCs revealed differences between the two groups (Figure 1a). The distribution of log2-fold change and *p*-value for the 12,347 genes expressed in these samples is shown in a volcano plot, with 207 DEGs (fold change > 2 or <0.5) highlighted in color (Figure 1b). A list of DEGs is provided in Appendix A. The heat map of the 207 DEGs, with 127 downregulated and 80 upregulated genes in the PBMCs of patients with IPF compared to the HCs, showed differences in transcriptome signatures between the two groups (Figure 1c). A heatmap with detailed gene names is shown in Appendix A.

Enrichment analysis was performed using DEGs to further explore the transcriptome signature in the PBMCs of patients with IPF compared to HCs. Gene ontology (GO) analysis revealed that various biological processes and molecular functions were enriched (Table 2a,b). The detailed results of the GO analyses of biological processes, molecular functions, and cellular components are shown in Appendix A. KEGG pathway analysis revealed that multiple pathways, including “Synaptic vesicle cycle (DEGs; RAB3A and STX1A)” and “Insulin secretion (DEGs; *RAB3A* and *STX1A*)”, were upregulated, and “Calcium signaling pathway (DEGs: *EGF*, *TNNC2*, *PDGFB*, *CACNA1A*, *PDGFA*, and *AVPR1A*)”, and “Fatty acid elongation (DEGs; *ACOT7* and *ELOVL7*)” were downregulated (Table 3).

### 2.3. Comparison of Transcriptome Signature in PBMCs before and after PFD Administration

To explore the potential effects of PFD on the transcriptome signature of PBMCs in patients with IPF, RNA-seq libraries isolated from PBMCs were prepared before and after PFD administration (*n* = 3). The RNA integrity values of all the samples were >9, and 26,472 mRNA genes were identified. Following quality control, 11,962 genes were identified.

PCA comparing gene expression levels using reads per million (RPM) data before and after PFD administration revealed differences between the two groups (Figure 2a). The distribution of log2-fold change and *p*-value for the 11,962 genes expressed in these samples is shown in volcano plots, with 170 DEGs (fold change > 2 or <0.5) highlighted in color (Figure 2b). A detailed list of DEGs is presented in Appendix A. The heat map of the 170 DEGs, with 98 downregulated and 72 upregulated genes in PBMCs after PFD administration compared to those before PFD administration, showed modulation of the transcriptome signature by PFD treatment (Figure 2c). A heat map with detailed gene names is shown in Appendix A. Transcriptome levels of TGF-β1, TNF-α, and IL-1B, the known molecules related to potential PFD-acting mechanisms, were evaluated, and no significant differences were observed between those before and after PFD administration (Appendix A). *CCNE1* and *SERPINE1* expression levels before and after PFD administration were compared by real-time quantitative PCR. The results show that *SERPINE1* expression tended to be higher after PFD administration, supporting the results of transcriptome analysis using RNA sequencing (Appendix A).

To further explore the transcriptome signature modulated in PBMCs following PFD treatment, enrichment analysis was performed using the DEGs. GO analysis revealed that various biological processes and molecular functions were enriched (Table 4a,b). The detailed results of the GO analyses of biological processes, molecular functions, and cellular components are shown in Appendix A. KEGG pathway analysis revealed that “p53 signaling pathway (DEGs; *CCNE1* and SERPINE1)” was upregulated, while “Amoebiasis (DEGs; *COL1A1*, *GNAL*, and *C8G*)” was downregulated (Table 5).

### 2.4. Comparison of Transcriptome Signature in PBMCs before and after NTD Administration

To explore the potential effects of the anti-fibrotic drug NTD on the transcriptome signature of PBMCs from patients with IPF, RNA-seq libraries isolated from PBMCs were prepared before and after NTD administration (n = 3). The RNA integrity values of all the samples were >9, and 26,472 mRNA genes were identified. After additional quality control, 12,123 genes were identified.

PCA comparing gene expression levels using RPM data between the patients before and after NTD administration revealed that the two groups differed (Figure 3a). The distribution of log2-fold change and *p*-value for the 12,123 genes expressed in these samples are shown in volcano plots, with 198 DEGs (fold change > 2 or <0.5) highlighted in color (Figure 3b). A detailed list of the DEGs is presented in Appendix A. The heat map of the 198 DEGs, with 103 downregulated and 95 upregulated genes in PBMCs after NTD administration compared with those before NTD administration, showed modulation of the transcriptome signature by NTD treatment (Figure 3c). A heat map with detailed gene names is shown in Appendix A. The transcriptome levels of VEGFRs, PDGFRs, and FGFRs, which are known molecules related to potential NTD mechanisms, were evaluated, and no significant differences were observed before and after NTD administration (Appendix A). Additionally, the expression levels of *ACOT7, CCNB2, CDK1,* and *SERPINE1* before and after NTD administration were compared using real-time quantitative PCR. The results show that the expression levels of these genes tended to be higher after NTD administration, supporting the results of transcriptome analysis using RNA sequencing (Appendix A).

To further explore the transcriptome signature modulated in PBMCs following NTD treatment, enrichment analysis was performed using DEGs. GO analysis revealed that various biological processes and molecular functions were enriched (Table 6a,b). The results in detail of the GO analyses of biological processes, molecular functions, and cellular components are shown in Appendix A. KEGG pathway analysis revealed that pathways that included “Fatty acid elongation (DEGs; *ACOT7*, *ELOVL6*, and *PPT2*)” and “p53 signaling pathway (DEGs; *CCNB2*, *SERPINE1*, and *CDK1*)” were upregulated and “Insulin secretion (DEGs; *KCNN3* and *CACNA1F*)” was downregulated (Table 7).

## 3. Discussion

In the present study, transcriptome analysis of bulk PBMCs showed that gene expression profiles differed between patients with IPF and HCs and were modified by three months of anti-fibrotic drug treatment. GO and KEGG pathway enrichment analyses revealed that various biological processes and pathways related to IPF pathogenesis were enriched (Table 2 and Table 3). PBMCs, including monocytes and lymphocytes, are considered immunocompetent cells. We speculated that the transcriptome of bulk PBMCs reflects certain aspects of the pathobiology of the final lung reconstruction. IPF is a pathological condition caused by an abnormal lung repair process, which leads to myofibroblast proliferation and ECM accumulation in the lungs. A recent and growing body of literature has described the key role of immune cells in orchestrating cell–cell interactions, including the crosstalk between epithelial cells and fibroblasts [9,10,11]. We detected different DEGs between the PFD and NTD treatment groups (Table 4, Table 5, Table 6 and Table 7); therefore, our study potentially indicated the different mechanistic roles of the two anti-fibrotic drugs.

This study embraced a novel approach by using RNA sequencing to examine PBMCs in IPF, potentially revealing systemic biomarkers or pathways that could be targeted for therapy. Herazo-Maya et al. have hypothesized that PBMC gene expression patterns using microarray analyses may be predictive of poor outcomes in patients with IPF, demonstrating that decreased expression of genes belonging to “The costimulatory signal during T cell activation,” in particular CD28, is associated with shorter survival [12]. Fraser E et al. have reported that monocytes from patients with IPF displayed increased expression of CD64 (FcγR1), which correlated with the amount of lung fibrosis and an amplified type I interferon response ex vivo, indicating that a primed type I interferon pathway may contribute to driving chronic inflammation and fibrosis [14].

*RAB3A* and *STX1A*, included in the “synaptic vesicle cycle,” were upregulated in patients with IPF compared with HCs in both the GO biological process and KEGG pathway (Table 2 and Table 3). The Ras-related protein Rab-3A is encoded by *RAB3A*, a member of the RAS oncogene family in humans. Epithelial-to-mesenchymal transition (EMT) is indispensable for epithelial cells to confer migratory and invasive properties to maintain phenotypic plasticity during lung repair in progressive fibrosis [15,16]. TGF-β is a potent inducer of fibrogenic EMTs; aberrant TGF-β signaling and EMTs are implicated in the pathogenesis of pulmonary fibrosis [17]. TGF-β depends on RAS and mitogen-activated protein kinase pathway inputs to induce EMTs [18]. Su et al. demonstrated that RAS-responsive element-binding protein 1, a RAS transcriptional effector, is a key partner of TGF-β-activated Smad transcription factors in EMT, suggesting that the RAS oncogene family could play an important role in the regulation of epithelial plasticity and its pathophysiological consequences in lung fibrosis [19]. Therefore, *RAB3A* may participate in EMT progression through TGF-β signaling in lung fibrosis. Syntaxin 1A (brain) (STX1A) is a t-SNARE protein involved in the fusion of synaptic vesicles and neurotransmitter release. *STX1A* is also expressed in airway epithelial cells [20] and negatively modulates *CFTR* function by altering intracellular trafficking and/or channel activity [21]. The “calcium signaling pathway” was downregulated in patients with IPF in KEGG pathway analysis (Table 3). In the airway epithelium of the lung, Ca-stimulated rapid secretion of mucin produces highly sticky mucus that adheres to the airway wall and causes airway obstruction and infection, contributing to IPF deterioration [22,23,24]. Negative modulation of *CFTR* function and downregulation of the “calcium signaling pathway” might contribute to avoiding the worsening of IPF, although it remains unknown how *STX1A* upregulation modulates IPF pathophysiology.

Fermitin family homolog 2 (FERMT2) is also known as pleckstrin homology domain-containing family C member 1 (PLEKHC1) or kindlin-2. *FERMT2*, which is involved in “adherens junction maintenance,” “regulation of wound healing, spreading of epidermal cells,” and “positive regulation of integrin activation,” was upregulated in patients with IPF compared to HCs (Table 2). According to GO: molecular functions, *FERMT2* was also upregulated in “Type I Transforming Growth Factor Beta Receptor Binding.” In addition, *FERMT2* was downregulated by PFD treatment according to GO: biological process (Table 4a) and GO: molecular function (Table 4b). Kindlin-2 and its binding partner, PYCR1, a key enzyme in proline synthesis, are increased in fibrotic human lung tissues. Zhang et al. demonstrated that TGF-β1 treatment increased the expression of kindlin-2 and PYCR1 in human lung fibroblasts, resulting in increased kindlin-2 mitochondrial translocation, formation of the kindlin-2-PYCR1 complex, and proline synthesis. Furthermore, the kindlin-2-PYCR1 complex and proline synthesis were reduced in response to PFD and NTD [25]. Therefore, kindlin-2 may be a key promoter of lung fibroblast activation and collagen matrix synthesis in pulmonary fibrosis. “Adherens junction maintenance” GO was upregulated in the IPF group (Table 2) and downregulated after PFD administration (Table 4). The expression levels of adherence junction proteins are increased in the alveolar epithelium of patients with pulmonary fibrosis compared to those in normal alveolar epithelium [26].

*ELOVL7*, which is included in three GO terms, “fatty acid elongation, monounsaturated fatty acid,” “polyunsaturated fatty acid,” and “unsaturated fatty acid,” were downregulated in patients with IPF compared with HCs (Table 2). *ELOVL6*, which was incorporated into the same GO term as *ELOVL7*, was upregulated by the NTD treatment (Table 6). Furthermore, *ELOVL6* is included in “fatty acid elongation” in the KEGG pathway (Table 7). Collectively, a decrease in fatty acid elongation could play a mechanistic role in IPF pathogenesis, and, in this respect, NTD might influence the decrease in fatty acid elongation recovery. Under physiological conditions, the elongation enzyme ELOVL6 allows fatty acid elongation, promoting the increase of intracellular stearic acid, which interferes with the profibrotic TGF-β/Smad signaling. During pulmonary fibrosis, *ELOVL6* expression decreases, favoring palmitic acid accumulation. Increased intracellular palmitic acid induces oxidative stress, thus promoting TGF-β/Smad signaling and cell activation. This effect is enhanced by the accumulation of cholesterol and its derivatives during fibrosis, which activate the expression of collagen and other ECM components. In addition, surfactant lipids accumulate within the cells because of the decreased expression of ABCA3 under fibrotic conditions. This impairs surfactant formation during pulmonary fibrosis [27,28]. The upregulated *ELOVL6* by NTD administration in the present study exhibited anti-fibrotic effects by suppressing TGF-β/Smad signaling and oxidative stress production. Therefore, alterations in fatty acid metabolism were associated with fibrosis via apoptosis, epithelial–mesenchymal transition, and endoplasmic reticulum stress during fibroblast differentiation. Moreover, some fatty acids are agonists of peroxisome proliferator-responsive receptors that can inhibit the TGF-β action [29,30]. NTD may affect fatty acid metabolism in PBMCs and inhibit the progression of pulmonary fibrosis.

“Negative regulation of blood coagulation,” in which *PDGFB* and *PDGFA* are included, was downregulated in patients with IPF (Table 2), whereas “negative regulation of plasminogen activation”, in which *SERPINE1* is included, was upregulated by PFD treatment (Table 4) and “negative regulation of blood coagulation” and “negative regulation of plasminogen activation,” in which *SERPINE1* and *SERPINE2* are included, was upregulated by NTD treatment (Table 6). Excessive coagulation and protease-activated receptor (PAR) signaling may contribute to inflammatory responses and lung fibrosis [31]. This study suggests that the fibrinolytic coagulation system is associated with the pathogenic mechanisms involved in the development of lung fibrosis. Although the pharmaceutical interview form for PFD followed the example of conventional angiogenesis inhibitors (Bevacizumab, VEGF receptor blocker) and described adverse events according to the description of bevacizumab, no serious side effects regarding bleeding tendency or thrombosis were reported with PFD.

*SERPINE1* encodes a protein. Diseases associated with *SERPINE1* include plasminogen activator inhibitor-1 (PAI-1) deficiency. PAI-1 regulates wound healing by inhibiting plasmin-mediated matrix metalloproteinase activation and inhibition of PAI-1 activity attenuates lung fibrosis. Disinhibition of plasmin-mediated matrix metalloproteinase activation leads to collagen degradation and diminishes its accumulation, resulting in reduced deposition of the fibrotic matrix in the lungs. In this study, PFD and NTD treatments upregulated *SERPINE1* expression, reflecting the anti-fibrotic effects of these two drugs. *SERPINE2* encodes a serine protease inhibitor with activity toward thrombin, trypsin, and urokinase, although its role in lung fibrosis has not yet been defined [32]. PDGF is a potent mitogen for fibroblasts [33] and appears to play an essential role in myofibroblast expansion by stimulating proliferation and migration [34]. PDGF is mainly produced by alveolar macrophages and epithelial cells in human lungs [34,35]. In this study, *PDGFB* and *PDGFA*, included in the GO term “negative regulation of blood coagulation,” were downregulated in IPF, although PDGF gene networks should be activated in patients with IPF. Platelet-derived PDGF may also play a role in wound healing. Platelets promote angiogenesis, and substances stored in their granules mediate this effect. It has been suggested that PDGF is mainly effective during the migration stage, whereas the combined effects of basic fibroblast growth factor and VEGF are crucial for vessel formation [36]. The role of PDGF differs depending on disease stage.

KEGG pathway analysis revealed that the “p53 signaling pathway” was upregulated by PFD or NTD treatment (Table 5 and Table 7); however, it was not present in the DEGs between patients with IPF and HCs. When DNA damage in epithelial cells is caused by injury, wild-type p53 levels increase, resulting in cell cycle arrest in the G1 phase or apoptosis. In the p53 signaling pathway, p21 protein expression is induced by p53 activation, which suppresses *CCNE1*. p53 accelerates the onset of pulmonary fibrosis by promoting the apoptosis of alveolar epithelial cells and the progression of EMT [37,38]. In contrast, increased p53 levels in lung fibroblasts promote apoptosis and delay the onset of pulmonary fibrosis [39]. The present study suggests that the p53 signaling pathway is associated with the pathogenesis of IPF, and that PFD and NTD treatments may influence the p53 signaling pathway at the PBMC level, ultimately leading to suppressive effects on pulmonary fibrosis progression.

GO analysis revealed that “positive regulation of integrin activation” was upregulated in the IPF group (Table 2) and was downregulated after PFD administration (Table 4). Furthermore, “negative regulation of integrin-mediated cell adhesion” was upregulated by PFD or NTD treatment (Table 4 and Table 6). Integrins are cell adhesion and signaling proteins that are crucial for a wide range of biological functions. Effectively targeted integrins differ in pathologies, such as fibrotic lung diseases, nonalcoholic steatohepatitis, and cardiovascular diseases. α_v_ integrins, key regulators of TGF-β activation and fibrogenesis in the in vivo models of pulmonary fibrosis, are expressed on abnormal epithelial cells (α_v_β_6_) and fibroblasts (α_v_β_1_) in fibrotic lungs [40]. Integrins are related to fibrosis progression by affecting TGF-β activation and downstream signaling [41]. PFD decreases the expression of profibrotic factors such as TGF-β. In this study, anti-fibrotic treatment with PFD or NTD affected integrin function at the PBMC level, suggesting the suppression of pulmonary fibrosis progression.

GO analysis revealed that “regulation of wound healing” was upregulated in the IPF group (Table 2) and downregulated by PFD or NTD treatment (Table 4 and Table 6). IPF is caused by repeated abnormal wound healing in the alveolar epithelium and structural lung remodeling by the eventual replacement of the lung parenchyma with ECM deposition [42]. In this study, NTD downregulated *F2RL1*, which encodes PAR2, coagulation factor II (thrombin) receptor-like 1 and G protein-coupled receptor 11. PAR2 is expressed in several types of immune cells and modulates the inflammatory response. Uncontrolled coagulation and PAR signaling responses have been shown to contribute to excessive inflammatory and fibroproliferative responses in fibrotic lung disease. Coagulation zymogens are thought to be derived from circulatory immune cells and are locally activated via the extrinsic tissue factor-dependent coagulation pathway within the intra-alveolar compartment. PAR1 influences endothelial–epithelial barrier disruption, inflammatory cell recruitment, and collagen deposition in response to acute lung injury, whereas PAR2 signaling has been implicated in mediating lung inflammatory responses [31]. Therefore, PFD plays a partial anti-fibrotic role by suppressing the coagulation pathway.

This study revealed downregulation of *COL1A1* following PFD treatment (Table 5). Immunocompetent cells in the lungs, including macrophages and monocytes, orchestrate the progression and maintenance of fibrosis. Collagen expression plays a novel role in monocyte-derived macrophages during tissue damage and wound healing. Tsitoura et al. reported higher COL1A1 levels in the bronchoalveolar lavage fluid of patients with a usual interstitial pneumonia pattern and a progressive fibrosing phenotype. Macrophages isolated from bronchoalveolar lavage and stained with CD45 contain higher levels of COL1A1, suggesting that profibrotic airway macrophages are increased in patients with IPF [43]. PFD suppressed activated macrophage function.

*PLA2G4A* (phospholipase A2 group IVA), *RNLS* (renalase, an FAD-dependent amine oxidase), and *HLA-DOA* (major histocompatibility complex class II DO alpha) were downregulated by NTD treatment, although they were not upregulated in patients with IPF compared with HCs. Cytosolic phospholipase A(2) (cPLA(2)) is related to the production of thromboxanes and leukotrienes in the lung. Nagase et al. demonstrated that cPLA(2)-downregulated mice with a disrupted *PLA2G4A* gene presented attenuated lung inflammation and fibrosis induced by bleomycin administration, suggesting that cPLA(2) is a key enzyme in the generation of proinflammatory eicosanoids in pulmonary fibrosis [44]. During the exacerbation of IPF, one imagines that any trigger activates immunocompetent cells, including monocytes (macrophages), and a robust surge in proinflammatory cytokines is induced, followed by the release of reactive oxygen species, which synergistically results in a fibroproliferative lung response. Systemic administration of renalase (a novel amino oxidase) alleviated experimentally induced lung fibrosis, suggesting that it exploits cytoprotective mechanisms, favoring cellular protection [45]. If NTD downregulates *RNLS*, it may interfere with its anti-fibrotic effects. Lung emphysema is an abnormal airspace enlargement followed by destruction of the lung structure without any prominent fibrosis. From this perspective, one might say that during lung emphysema formation, any anti-fibrotic signal acts to prevent lung fibrosis. *HLA-DOA* is listed as a crucial gene related to molecular pathogenesis in activated pathways, such as the immune response IL-1 signaling pathway, positive regulation of smooth muscle migration, BMP signaling pathway, positive regulation of leukocyte migration, NF-kappa B signaling, and cytochrome-c oxidase activity [46].

*FEN1* (Flap Structure-Specific Endonuclease 1), *CDC20* (cell division cycle 20), and *FERMT2* were downregulated by PFD, whereas only *FERMT2* was upregulated in patients with IPF. Aging contributes to the establishment of IPF by altering the expression of genes related to mitochondrial function, cellular senescence, and telomeric length. *FEN1*, a gene associated with mitochondrial biogenesis and function, partially regulates the pathobiology of aging in IPF [47]. *CDC20* acts as a regulatory protein that interacts with several other proteins during multiple cell–cycle stages. In a murine model of asbestos-induced fibrogenesis, several biological processes, including inflammation, proliferation, and matrix remodeling, were shown to be involved in the development and repair of asbestos. *CDC20* is a candidate gene involved in cell proliferation [48].

This study has several limitations. First, we considered the possible mechanistic differences in the functions of these two anti-fibrotic drugs based on step-by-step assumptions. It was assumed that the DEG profiles resulted in the clinically meaningful expression of certain genes, as the qRT-PCR of *CCNE1* and *SERPINE1* after PFD treatment and *ACOT7, CCNB2, CDK1,* and *SERPINE1* after NTD treatment showed a similar trend, although not significant, to the transcriptome signature. Furthermore, we assumed that the overexpression of these genes in PBMCs would be relevant to pulmonary fibrosis in the same way as in pulmonary fibroblasts (e.g., *COL1A1* and *SERPINE1*). The results of the present study, which demonstrated that three months of anti-fibrotic treatment with PFD or NTD affected the RNA-seq analysis of PBMC, would have some pathobiological implications and suggest possible mechanistic differences in the functions of these two anti-fibrotic drugs. However, the comparison of effects between these two anti-fibrotic drugs was made indirectly, and we could not validate these results in a clinical setting.

Second, the association between the PBMC signature and IPF pathogenesis or the effects of anti-fibrotic treatment on the PBMC signature was based on the presupposition that immune cells in PBMCs are predisposed to those within the lungs and that they orchestrate cell–cell interactions, including crosstalk between epithelial cells and fibroblasts. Another premise is the effect of anti-fibrotic drug administration on the inflammatory aspects of the PBMC signature. Overcoming these assumptions, this study suggests that DEGs in the PBMC signature partly reflect the proposed process.

Third, this study was conducted at a single center in Japan, and the sample size was small (n = 6 for IPF and n = 6 for healthy controls; n = 3 for each drug treatment), potentially limiting the statistical power and the ability to generalize findings or identify less robust differentially expressed genes in this study, since IPF is a cluster of clinical phenotypes. Additionally, the false discovery rate of the q-value should be taken into consideration when interpreting the *p*-values (Appendix A). Meanwhile, the raw RPM data of the representative gene expression levels are shown in Appendix A and suggest that results of this study could be supported if one random patient from the correspondent group is excluded.

Fourth, the genetic abnormalities may differ among ethnic groups. Specific genetic abnormalities and polymorphisms have been identified in some cases of familial IPF and in some cases of patients with sporadic IPF [49,50]. In the present study, all the participants in were Asian and male, thus careful interpretation of the transcriptional signatures is necessary to account for ethnicity and sex.

Fifth, the composition of PBMC might be different among participants, as it contains monocytes (with 3 canonical subsets) and lymphocytes (different subsets of B cells and CD4 or CD8 T cells) and possibly a small percentage of granulocytes. The differences in composition, rather than up/downregulation of gene expression, may account for the changes observed. Therefore, further research would be required to focus on the possible differences in isolated leucocyte populations among participants and changes after treatment.

Finally, our study design did not include long-term follow-up, and the three-month treatment period may not be sufficient to reflect all the transcriptomic changes related to treatment. Therefore, further research is necessary to confirm whether similar results are observed with longer follow-up.

## 4. Materials and Methods

### 4.1. Participants

The Human Ethics Committee of Chiba University approved this study (protocol no. 2083). PBMCs were collected from patients diagnosed with IPF and HCs between October 2020 and February 2023 at Chiba University Hospital. For patients with IPF, PBMCs were collected before and approximately three months after initiating treatment with anti-fibrotic drugs. IPF was diagnosed according to the 2018 ATS/ERS/JRS/ALAT IPF guidelines [2], and patients with malignant complications were excluded.

### 4.2. Isolation of PBMCs

Peripheral blood was collected in a BD Vacutainer CPT Cell Preparation Tube with sodium citrate, according to the manufacturer’s protocol (#362760) (Becton, Dickinson and Company, NJ, USA). Briefly, tubes containing blood were centrifuged at 1500 rpm at room temperature for 20 min. After centrifugation, plasma was removed from the uppermost layer. The PBMC layer was washed twice with phosphate-buffered saline, Isogen (Nippongene, Tokyo, Japan) added, and stored at −80 °C.

### 4.3. Total RNA Extraction, mRNA Library Preparation, and 3′RNA-seq

Total RNA extracted from 1.0–2.0 × 10^6^ PBMCs using 1.0 mL of Isogen reagent (Life Technologies, Carlsbad, CA, USA) was centrifuged after adding chloroform, and the aqueous phase was carefully transferred to a new tube, after which 10 mg of glycogen (Life Technologies) was added as a co-precipitant. RNA was precipitated by adding 600 μL of isopropyl alcohol. The RNA pellet was washed once using 75% ethanol and dissolved in 10 μL RNase-free water. The concentration and quality of the RNA were verified using a Qubit fluorometer (Life Technologies) and an Agilent 2100 bioanalyzer, respectively. Purified total RNA (200 ng) was used for RNA library preparation, according to the instructions of the Quant Seq 3′ mRNA-seq library preparation kit FWD for Illumina (Lexogen, Vienna, Austria). RNA libraries were sequenced on an Illumina NextSeq 500 system with 75-nt-long reads.

### 4.4. RNA-seq Data Analysis

Subsequently, 3′ RNA-seq data were processed using Perseus software (version 1.6.15.0; available online at https://maxquant.net/perseus/, accessed on 8 June 2023). The RPM data were log2-transformed and filtered to ensure that at least one group contained at least 70% of the valid values for each RNA. The remaining missing values were imputed using random numbers drawn from a normal distribution (width: ¼ 0.3; downshift: ¼ 2.8). Hierarchical clustering and heat maps were created using Perseus-processed data in Qlucore Omics Exploration software ver.3.9.9 (Qlucore AB, Lund, Sweden). DEGs were detected for each gene in both samples. The fold-changes between groups were >2 (upregulated) or <0.5 (downregulated) (*p* < 0.05). The false discovery rate was also calculated as a q-value and taken into consideration to interpret the *p*-values. Overrepresented functional categories were identified using Enrichr software (http://amp.pharm.mssm.edu/Enrichir/, accessed on 9 June 2023). Genes significantly upregulated or downregulated in “IPF vs. HCs” or “before vs. after anti-fibrotic drugs (PFD or NTD) administration” were annotated. Subsequently, GO terms and KEGG pathways were identified. The selected GO terms and KEGG pathways were considered statistically significant at *p* < 0.05.

### 4.5. Real-Time Quantitative PCR Analysis

The extracted RNA was reverse-transcribed via PCR using the SuperScript VILO cDNA Synthesis Kit (Thermo Fisher Scientific, Waltham, MA, USA) to synthesize single-stranded cDNA. The cDNA samples were amplified using predesigned probes in a duplex TaqMan real-time PCR system (Thermo Fisher Scientific). QuantStudio 12 K Flex was used for the analysis (Thermo Fisher Scientific). We performed real-time quantitative PCR on genes with a fold change greater than 4, and the endogenous control was GAPDH.

### 4.6. Statistical Analysis

The age characteristics of the samples are expressed as mean ± standard deviation. Statistical analyses were performed using JMP software version 16 (SAS Institute, Cary, NC, USA). Student’s *t*-test was used for age comparisons. Statistical significance was set at *p* < 0.05.

## 5. Conclusions

This study suggests that bulk gene expression patterns in PBMCs differ between patients with IPF and HCs and are affected by anti-fibrotic treatment of IPF. Changes in the gene expression patterns of PBMCs reflect the pathogenesis of IPF and the pharmacological effects of these two anti-fibrotic drugs. The results of the present study demonstrated that three months of anti-fibrotic treatment with PFD or NTD affected the RNA sequencing analysis of PBMC, had some pathobiological significance, and suggested possible mechanistic differences in the functions of these two anti-fibrotic drugs, although we could not prove the validity of these results in a clinical setting. These new findings may lead to a better understanding of IPF pathogenesis and the identification of potential therapeutic targets for IPF.

## Figures and Tables

**Figure 1 ijms-25-03750-f001:**
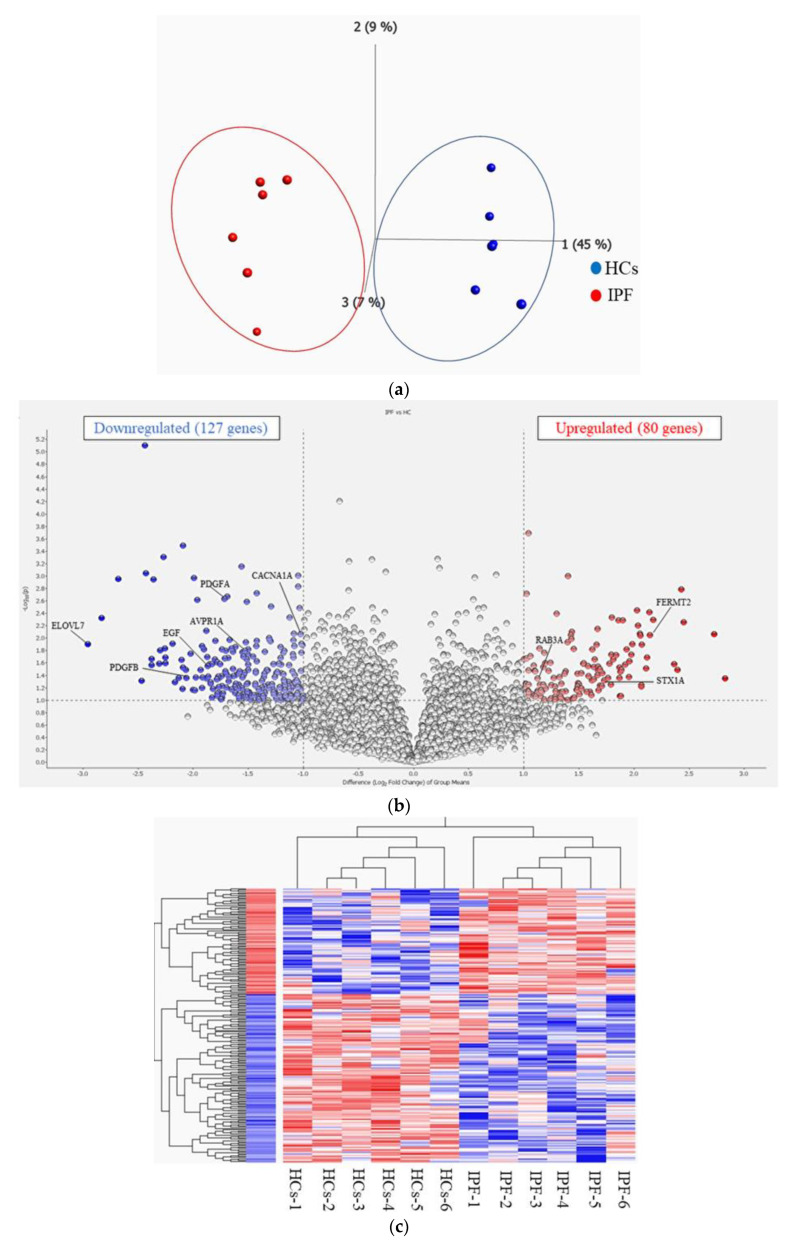
(**a**) Principal component analysis (PCA). PCA showing that two groups of idiopathic pulmonary fibrosis (IPF) and healthy controls (HCs) are well-differentiated. (**b**) Volcano plot of differentially expressed genes (DEGs). Volcano plot showing the distribution of log2-fold change and *p*-value for the 12,347 genes. Colored dots represent 207 DEGs between IPF and HCs. Red dots represent high expression, and blue dots represent low expression. (**c**). Heatmap of DEGs in IPF vs. HCs. Red bars represent high expression, and blue bars represent low expression. A heatmap with detailed gene names is shown in Appendix A.

**Figure 2 ijms-25-03750-f002:**
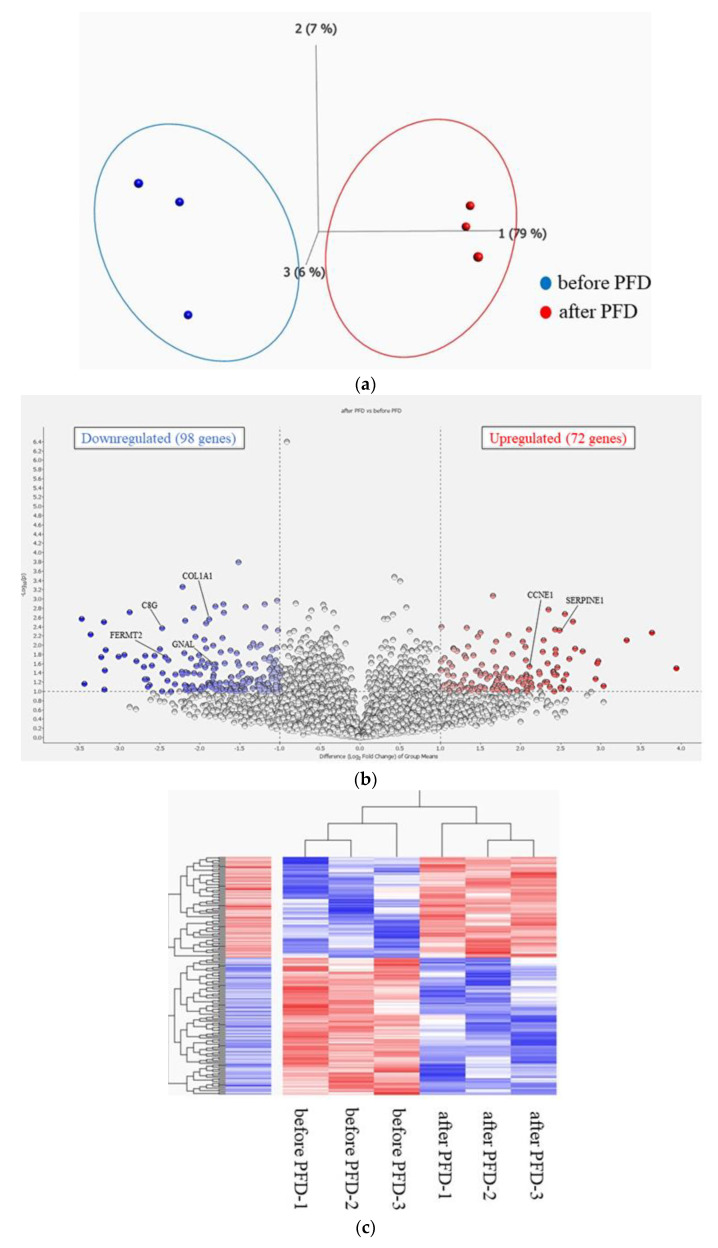
(**a**) Principal component analysis (PCA). PCA showing that the two groups before and after PFD administration are well-differentiated. (**b**) Volcano plot of DEGs. Volcano plot showing the distribution of log2-fold change and *p*-value for the 11,962 genes. Colored dots represent 170 DEGs between before and after PFD administration. Red dots represent high expression, and blue dots represent low expression. (**c**) Heatmap of differentially expressed genes (DEGs). Heatmap showing the DEGs before and after PFD administration. Red bars represent high expression, and blue bars represent low expression. A heatmap with detailed gene names is shown in Appendix A.

**Figure 3 ijms-25-03750-f003:**
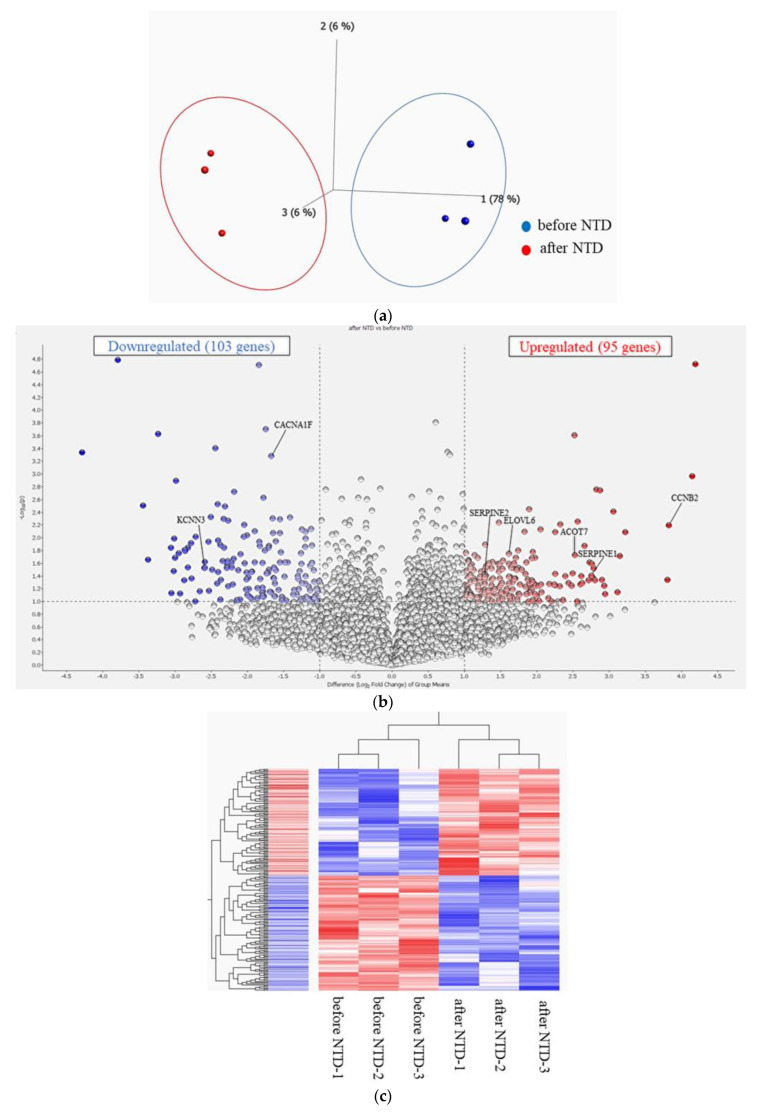
(**a**) Principal component analysis (PCA). PCA showing that the two groups before and after NTD administration are well-differentiated. (**b**) Volcano plot of DEGs. Volcano plot showing the distribution of log2-fold change and *p*-value for the 12,123 genes. Colored dots represent 198 DEGs between before and after NTD administration. Red dots represent high expression, and blue dots represent low expression. (**c**) Heatmap of DEGs. Heatmap showing the DEGs before and after NTD administration. Red bars represent high expression, and blue bars represent low expression. A heatmap with detailed gene names is shown in Appendix A.

**Table 1 ijms-25-03750-t001:** Participant characteristics and the changes in FVC 3 months after PFD and NTD.

Idiopathic pulmonary fibrosis (IPF)							
Samples	Sex	Age	Smoking history	Complications	Anti-fibrotic drugs	FVC (L) (Before anti-fibrotic drugs)	FVC (L) (Three months after anti-fibrotic drugs)
No. 1	male	76	Yes	HT, DL, DM	PFD	2.19	2.04
No. 2	male	74	Yes	DL	PFD	2.27	2.36
No. 3	male	68	Yes	None	PFD	1.82	1.82
No. 4	male	77	Yes	None	NTD	2.56	2.77
No. 5	male	66	Yes	HT	NTD	2.53	2.88
No. 6	male	66	Yes	HT, DL	NTD	2.50	2.31
Healthy controls (HCs)							
Samples	Sex	Age	Smoking history	Complications			
No. 1	male	70	None	None			
No. 2	male	67	None	DM			
No. 3	male	67	None	HT, DM			
No. 4	male	71	Yes	hay fever			
No. 5	male	63	Yes	HT, DL			
No. 6	male	81	Yes	HT			
IPF vs. HCs							
	IPF		HCs	*p*-value			
Age	71.2 ± 5.1		69.8 ± 6.1	0.69			

HT, hypertension; DL, dyslipidemia; DM, diabetes mellitus; PFD, pirfenidone; NTD, nintedanib; FVC, forced vital capacity.

**Table 2 ijms-25-03750-t002:** (**a**) Enrichment analysis of transcriptome data. Gene Ontology (biological process): IPF vs. HCs. Relevant terms were excerpted. (**b**) Enrichment analysis of transcriptome data. Gene Ontology (molecular functions): IPF vs. HCs. Relevant terms were excerpted.

(**a**)		
Terms (Gene Ontology: biological process) with upregulated genes	*p*-value	DEGs
Synaptic vesicle cycle (GO:0099504)	0.017	*RAB3A, STX1A*
Adherens junction maintenance (GO:0034334)	0.023	*FERMT2*
Regulation of wound healing, spreading of epidermal cells (GO:1903689)	0.027	*FERMT2*
Positive regulation of integrin activation (GO:0033625)	0.039	*FERMT2*
Terms (Gene Ontology: biological process) with downregulated genes	*p*-value	DEGs
Negative regulation of blood coagulation (GO:0030195)	0.015	*PDGFB*, *PDGFA*
Fatty acid elongation, monounsaturated fatty acid (GO:0034625)	0.037	*ELOVL7*
Fatty acid elongation, polyunsaturated fatty acid (GO:0034626)	0.037	*ELOVL7*
Fatty acid elongation, unsaturated fatty acid (GO:0019368)	0.037	*ELOVL7*
(**b**)		
Terms (Gene Ontology: molecular functions) with upregulated genes	*p*-value	DEGs
Type I Transforming Growth Factor Beta Receptor Binding (GO:0034713)	0.035	*FERMT2*

**Table 3 ijms-25-03750-t003:** Enrichment analysis of transcriptome data. KEGG pathway: IPF vs. HCs. Relevant terms were excerpted.

Terms (KEGG pathway) with upregulated genes	*p*-value	DEGs
Synaptic vesicle cycle	0.039	*RAB3A*, *STX1A*
Insulin secretion	0.046	*RAB3A*, *STX1A*
Terms (KEGG pathway) with downregulated genes	*p*-value	DEGs
Calcium signaling pathway	0.004	*EGF*, *TNNC2*, *PDGFB*, *CACNA1A*, *PDGFA*, *AVPR1A*
Fatty acid elongation	0.012	*ACOT7*, *ELOVL7*

**Table 4 ijms-25-03750-t004:** (**a**) Enrichment analysis of transcriptome data. Gene Ontology (biological process): before vs. after PFD administration. Relevant terms were excerpted. (**b**) Enrichment analysis of transcriptome data. Gene Ontology (molecular functions): before vs. after PFD administration. Relevant terms were excerpted.

(**a**)		
Terms (Gene Ontology: biological process) with upregulated genes	*p*-value	DEGs
Negative regulation of plasminogen activation (GO:0010757)	0.021	*SERPINE1*
Negative regulation of cell adhesion mediated by integrin (GO:0033629)	0.031	*SERPINE1*
Terms (Gene Ontology: biological process) with downregulated genes	*p*-value	DEGs
Adherens junction maintenance (GO:0034334)	0.029	*FERMT2*
Regulation of wound healing, spreading of epidermal cells (GO:1903689)	0.037	*FERMT2*
Positive regulation of integrin activation (GO:0033625)	0.038	*FERMT2*
(**b**)		
Terms (Gene Ontology: molecular functions) with downregulated genes	*p*-value	DEGs
RNA Endonuclease Activity, Producing 5′-Phosphomonoesters (GO:0016891)	0.008	*FEN1*
Phosphatidylinositol-3,4,5-Trisphosphate Binding (GO:0005547)	0.013	*FERMT2*
Phosphatidylinositol Phosphate Binding (GO:1901981)	0.016	*FERMT2*
5′-Flap Endonuclease Activity (GO:0017108)	0.024	*FEN1*
Flap Endonuclease Activity (GO:0048256)	0.029	*FEN1*
Ubiquitin Ligase Activator Activity (GO:1990757)	0.029	*CDC20*
Endodeoxyribonuclease Activity, Producing 5′-Phosphomonoesters (GO:0016888)	0.039	*FEN1*
Type I Transforming Growth Factor Beta Receptor Binding (GO:0034713)	0.043	*FERMT2*

**Table 5 ijms-25-03750-t005:** Enrichment analysis of transcriptome data. KEGG pathway: before vs. after PFD administration. Relevant terms were excerpted.

Terms (KEGG Pathway) with upregulated genes	*p*-value	DEGs
p53 signaling pathway	0.028	*CCNE1*, *SERPINE1*
Terms (KEGG Pathway) with downregulated genes	*p*-value	DEGs
Amoebiasis	0.013	*COL1A1*, *GNAL*, *C8G*

**Table 6 ijms-25-03750-t006:** (**a**) Enrichment analysis of transcriptome data. Gene Ontology (biological process): before vs. after NTD administration. Relevant terms were excerpted. (**b**) Enrichment analysis of transcriptome data. Gene Ontology (molecular functions): before vs. after NTD administration. Relevant terms were excerpted.

(**a**)		
Terms (Gene Ontology: biological process) with upregulated genes	*p*-value	DEGs
Negative regulation of plasminogen activation (GO:0010757)	0.0003	*SERPINE1*, *SERPINE2*
Negative regulation of blood coagulation (GO:0030195)	0.008	*SERPINE1*, *SERPINE2*
Fatty acid elongation, monounsaturated fatty acid (GO:0034625)	0.028	*ELOVL6*
Fatty acid elongation, polyunsaturated fatty acid (GO:0034626)	0.028	*ELOVL6*
Fatty acid elongation, unsaturated fatty acid (GO:0019368)	0.028	*ELOVL6*
Negative regulation of cell adhesion mediated by integrin (GO:0033629)	0.041	*SERPINE1*
Terms (Gene Ontology: biological process) with downregulated genes	*p*-value	DEGs
Regulation of wound healing (GO:0061041)	0.024	*KANK1*, *F2RL1*
(**b**)		
Terms (Gene Ontology: molecular functions) with downregulated genes	*p*-value	DEGs
Ceramide 1-Phosphate Binding (GO:1902387)	0.026	*PLA2G4A*
Epinephrine Binding (GO:0051379)	0.026	*RNLS*
Oxidoreductase Activity, Acting On The CH-NH2 Group Of Donors, Oxygen As Acceptor (GO:0016641)	0.036	*RNLS*
Calcium-Independent Phospholipase A2 Activity (GO:0047499)	0.036	*PLA2G4A*
MHC Class II Receptor Activity (GO:0032395)	0.045	*HLA-DOA*

**Table 7 ijms-25-03750-t007:** Enrichment analysis of transcriptome data. KEGG pathway: before vs. after NTD administration. Relevant terms were excerpted.

Terms (KEGG Pathway) with upregulated genes	*p*-value	DEGs
Fatty acid elongation	0.0002	*ACOT7*, *ELOVL6*, *PPT2*
p53 signaling pathway	0.005	*CCNB2*, *SERPINE1*, *CDK1*
Terms (KEGG Pathway) with downregulated genes	*p*-value	DEGs
Insulin secretion	0.072	*KCNN3*, *CACNA1F*

## Data Availability

The datasets presented in this study can be found in online repositories. The names of the repository/repositories and accession number(s) can be found below: https://www.ncbi.nlm.nih.gov/, GSE248485.

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
