# Peer review of "Effects of Anti-Fibrotic Drugs on Transcriptome of Peripheral Blood Mononuclear Cells in Idiopathic Pulmonary Fibrosis"

_ijms, 2024, doi:10.3390/ijms25073750_

Round 1

Reviewer 1 Report (New Reviewer)

Comments and Suggestions for Authors

Daisuke Ishii et al report on the effects of Pirfenidone (PFD) and nintedanib (NTD) which are used to treat idiopathic pulmonary fibrosis (IPF) and show PBMCs play a role in IPF pathogenesis by orchestrating cell interactions in the lungs. RNA sequencing revealed differentially expressed genes (DEGs) affected by PFD and NTD. Enrichment analysis highlighted fatty acid elongation, TGF-β/Smad signaling, and PAI-1 regulation as key mechanisms.

The last sentence of the abstract reads "This study detected different DEGs between the PFD and NTD treatments, suggesting different mechanistic roles of the two anti-fibrotic drugs."  The comparisons were made indirectly.  A better way would be to calculate the fold change caused by each treatment and then compare the fold-changes for each treatment.  Alternatively, the authors could mention this limitation in the Discussion.

In several sections the authors repeat the list of GO terms which are listed in the Tables; this is not necessary.

When discussing the limitations of the study (starting at p15) the authors skip from third to fifth.  One limitation they should mention is that the composition of PMBC may be different between donors, as it contains monocytes (with 3 canonical subsets) and lymphocytes (different subsets of B cells and CD4 or CD8 T cells) and possibly a few % of granulocytes. The difference in composition, rather than up/down-regulation of gene expression may account for the changes observed - further research would be required to confirm changes in isolated leucocyte populations.

It would help the reader if the figure legends mentioned which supplementary figures are associated with each figure.

After refomatting, some figures could be combined; e.g. Figures 1 and 2,  Figures 4 and 5, Figures 7 and 8.

Comments on the Quality of English Language

English expression is fine.

Author Response

Reviewer 2 Report (New Reviewer)

Comments and Suggestions for Authors

Dear Colleagues,

Thank You for your efforts with this interesting work. Bellow please find few questions and comments 

1.      The study embraces a novel approach by using RNA-sequencing to examine PBMCs in IPF, potentially revealing systemic biomarkers or pathways that could be targeted for therapy. However extremely small sample size (n=6 for IPF and n=6 for healthy controls; n=3 for each drug treatment), may limit the statistical power and the ability to generalize findings or identify less robust differentially expressed genes. It will be good to estimate effect of simgle patient elimination on each particular results (e.g. estimate is each particular conclusion will be valid if we exclude one random patients from correspondent group)

2.      No mention is made of patient medication beyond the anti-fibrotic agents, which could affect gene expression profiles. This should be corrected

3.      Male is most affected sex for IPF. However, it will be good to specify study population in the title (e.g. “…  cells from male patients with Idiopathic Pulmonary Fibrosis”) or clearly mentioned in Limitation section.

4.      There is no mention of control for multiple testing (e.g., FDR correction) in the analysis of differential gene expression, which could lead to false positives

5.      Although transcriptomic alterations in PBMCs are hypothesized to reflect processes in lung tissue, direct evidence or correlations with lung pathology are not provided.

6.      The study design does not include long-term follow-up, and the three-month treatment period may not be sufficient to reflect all the transcriptomic changes related to treatment.

Round 2

Reviewer 1 Report (New Reviewer)

Comments and Suggestions for Authors

The authors have adequately addressed the points raised by reviewers in their revised document.

Please check the text for minor typographical errors e.g. Line 40 "interleukin (IL)-1" should be "interleukin (IL)-1β".

This manuscript is a resubmission of an earlier submission. The following is a list of the peer review reports and author responses from that submission.

Round 1

Reviewer 1 Report

Comments and Suggestions for Authors

In this study, Daisuke Ishii et al. describe a series of results obtained from a comprehensive RNA sequencing analysis of peripheral blood mononuclear cells from IPF patients, examining the effects of the antifibrotic drugs pirfenidone (PFD) and nintedanib (NTD) on these signatures. The analysis focused on differentially expressed genes (DEGs) in IPF patients and control subjects, both before and after antifibrotic treatment. It is essential to note that this study reveals significant divergences in DEGs resulting from PFD and NTD treatments, indicating possible mechanistic differences in the functions of these two antifibrotic drugs.

The authors have chosen a very relevant and important topic for their research. It is evident that they have invested a great deal of effort in the collection and analysis of the data. However, I have some minor observations regarding the submitted manuscript.

I would like to point out that a fundamental limitation of the study is the absence of validation of RNAseq results by real-time PCR technique. Validation through real-time PCR would be crucial to support and strengthen the robustness of the findings obtained by RNA sequencing. This additional validation would provide a layer of experimental confirmation, improving confidence in the results and their interpretation. It is suggested to consider the inclusion of this step to consolidate the reliability of the data presented.

A comprehensive presentation of a complete inventory of the differentially expressed genes (DEGs) identified in each of the comparison groups is suggested to the authors. It is proposed that these detailed tables be included as supplementary material appended to the manuscript, which would strengthen the transparency and accessibility of the data for further evaluation by the scientific community.

In the context of gene ontology (GO) analysis, the presentation of results related to biological processes (BP) in the manuscript is highlighted. However, it is suggested that the authors complement this analysis by including results pertaining to molecular functions (MF) and cellular components (CC). Incorporation of these additional aspects would enrich the comprehensive understanding of the biological implications of the findings, thus providing a more complete and detailed view of the functional dimensions associated with the identified gene expression.

It is suggested that the authors explicitly address the comparison of the differentially expressed genes (DEGs) and key genes identified in the present study with the results reported in previous research analyzing gene expression in the context of IPF. Inclusion of this comparison in the Discussion section would be of vital importance, as it would allow contextualization and assessment of the consistency of the current findings within the broader landscape of IPF research, thus providing a more complete and grounded perspective on the relevant scientific literature.

Overall, I congratulate the authors for their commendable efforts in their search for cures for disease and improvement of human life. I wish them success in improving their research.

Author Response

Response to the Comments of Reviewer 1

The reviewer raised several important points, thus the comments have been incorporated in the revised manuscript. We appreciate the thoughts and the time that the reviewer has given to this study. We hope the revisions are acceptable for publication in IJMS.

Comment 1

In this study, Daisuke Ishii et al. describe a series of results obtained from a comprehensive RNA sequencing analysis of peripheral blood mononuclear cells from IPF patients, examining the effects of the antifibrotic drugs pirfenidone (PFD) and nintedanib (NTD) on these signatures. The analysis focused on differentially expressed genes (DEGs) in IPF patients and control subjects, both before and after antifibrotic treatment. It is essential to note that this study reveals significant divergences in DEGs resulting from PFD and NTD treatments, indicating possible mechanistic differences in the functions of these two antifibrotic drugs. The authors have chosen a very relevant and important topic for their research. It is evident that they have invested a great deal of effort in the collection and analysis of the data. However, I have some minor observations regarding the submitted manuscript.

Response 1

We think that the reviewer pointed out very important points, including the limitation of this study. Therefore, we have revised our conclusion according to the reviewer’s kind suggestion.

In the revised manuscript

Line 548: Conclusions
This study suggests that bulk gene expression patterns in PBMCs differ between patients with IPF and HCs and are affected by antifibrotic treatment in IPF. Changes in the gene expression patterns of PBMCs reflect the pathogenesis of IPF and the pharmacological effects of the two antifibrotic drugs. The results of the present study demonstrated that three months of antifibrotic treatment with PFD or NTD affected RNA sequencing analysis of peripheral blood mononuclear cells, would have some pathobiological significance and suggested possible mechanistic differences in the functions of these two antifibrotic drugs, although we could not prove the validity of these results in a clinical setting. These new findings may lead to a better understanding of IPF pathogenesis and identify potential therapeutic targets for IPF.

Comment 2

I would like to point out that a fundamental limitation of the study is the absence of validation of RNAseq results by real-time PCR technique. Validation through real-time PCR would be crucial to support and strengthen the robustness of the findings obtained by RNA sequencing. This additional validation would provide a layer of experimental confirmation, improving confidence in the results and their interpretation. It is suggested to consider the inclusion of this step to consolidate the reliability of the data presented.

Response 2

We agreed with the review’s comment, since one of the limitations in the study is the absence of validation of our RNAseq results by real-time PCR technique. Our original question in this study was whether the effects of anti-fibrotic drugs appear in PBMC RNAseq, and whether the mechanistic roles of these drugs would be speculated from the immunocompetent cells. Our RNAseq results demonstrated some aspects of these drugs and should be validated by real-time PCR technique. We would like to perform real-time PCR in a next step, however in this manuscript we have added the matters related to PCR in the limitation of this study.

In the revised manuscript

Line 491: Third, in this study, we detected various DEGs in PBMCs altered by the intervention of antifibrotic drugs and identified candidate genes affected by the therapeutic drugs; however, these transcriptome changes require validation using real-time PCR.

Comment 3

A comprehensive presentation of a complete inventory of the differentially expressed genes (DEGs) identified in each of the comparison groups is suggested to the authors. It is proposed that these detailed tables be included as supplementary material appended to the manuscript, which would strengthen the transparency and accessibility of the data for further evaluation by the scientific community.

Response 3

We have agreed with the reviewer’s comment that we should show a complete inventory of the differentially expressed genes (DEGs) identified in each of the comparison groups. We have presented the lists of DEGs as Supplemental Tables as below.

Table S1: The details of differential gene expression in PBMCs between idiopathic pulmonary fibrosis (IPF) vs. healthy controls.
Table S5: The details of differential gene expression in PBMCs between before and after pirfenidone (PFD) administration.
Table S10: The details of differential gene expression in PBMCs between before and after nintedanib (NTD) administration.

Comment 4

In the context of gene ontology (GO) analysis, the presentation of results related to biological processes (BP) in the manuscript is highlighted. However, it is suggested that the authors complement this analysis by including results pertaining to molecular functions (MF) and cellular components (CC). Incorporation of these additional aspects would enrich the comprehensive understanding of the biological implications of the findings, thus providing a more complete and detailed view of the functional dimensions associated with the identified gene expression.

Response 4

In response to the reviewer’s comment, we have added the following tables in the text and presented supplementary tables.

In the revised manuscript

Table 2-2. Enrichment analysis of transcriptome data. Gene Ontology (molecular functions): idiopathic pulmonary fibrosis vs. healthy controls. Relevant terms were excerpted.
Table 4-2. Enrichment analysis of transcriptome data. Gene Ontology (molecular functions): before pirfenidone vs. after pirfenidone administration.
Table 6-2. Enrichment analysis of transcriptome data. Gene Ontology (molecular functions): before nintedanib vs. after nintedanib administration.
Table S2: The details of enrichment analysis of transcriptome data. Gene Ontology (biological process): idiopathic pulmonary fibrosis vs. healthy controls.
Table S3: The details of enrichment analysis of transcriptome data. Gene Ontology (molecular function): idiopathic pulmonary fibrosis vs. healthy controls.
Table S4: The details of enrichment analysis of transcriptome data. Gene Ontology (cellular component): idiopathic pulmonary fibrosis vs. healthy controls.
Table S7: The details of enrichment analysis of transcriptome data. Gene Ontology (biological process): before pirfenidone vs. after pirfenidone administration.
Table S8: The details of enrichment analysis of transcriptome data. Gene Ontology (molecular function): before pirfenidone vs. after pirfenidone administration.
Table S9: The details of enrichment analysis of transcriptome data. Gene Ontology (cellular component): before pirfenidone vs. after pirfenidone administration.
Table S12: The details of enrichment analysis of transcriptome data. Gene Ontology (biological process): before nintedanib vs. after nintedanib administration.
Table S13: The details of enrichment analysis of transcriptome data. Gene Ontology (molecular function): before nintedanib vs. after nintedanib administration.
Table S14: The details of enrichment analysis of transcriptome data. Gene Ontology (cellular component): before nintedanib vs. after nintedanib administration.

 In addition, we have added the related discussion to the manuscript.

In the discussion

Line 305: Fermitin family homolog 2 (FERMT2) is a protein also known as pleckstrin homology domain-containing family C member 1 (PLEKHC1) or kindlin-2. FERMT2, included in “adherens junction maintenance,” “regulation of wound healing, spreading of epidermal cells,” and “positive regulation of integrin activation,” was upregulated in patients with IPF compared with HCs (Table 2). According to GO: molecular functions, FERMT2 was also upregulated in “Type I Transforming Growth Factor Beta Receptor Binding.” In addition, FERMT2 was downregulated by PFD treatment according to GO: biological process (Table 4-1) and GO: molecular functions (Table 4-2).

Line 439: PLA2G4A (phospholipase A2 group IVA), RNLS (renalase, a FAD-dependent amine oxidase), and HLA-DOA (major histocompatibility complex class II DO alpha) were downregulated by NTB treatment, although they were not upregulated in patients with IPF compared with HCs. Cytosolic phospholipase A(2) (cPLA(2)) is related to the production of thromboxanes and leukotrienes in the lung. Nagase T et al. demonstrated that cPLA(2)-downregulated mice with a disrupted PLA2G4A gene presented an attenuated lung inflammation and fibrosis induced by bleomycin administration, thereby suggesting cPLA(2) in pulmonary fibrosis to be a key enzyme in the generation of proinflammatory eicosanoids[44]. In the exacerbation of IPF, one would imagine that any trigger activates immunocompetent cells, including monocytes (macrophages), and a robust surge in proinflammatory cytokines is induced, followed by the release of reactive oxygen species that synergistically results in a fibroproliferative lung response. Systemic renalase (a novel amino oxidase) administration alleviates experimentally induced lung fibrosis, suggesting that it exploits cytoprotective mechanisms, favoring cellular protection [45]. If NTB treatment downregulates RNLS, it may interfere with its antifibrotic effects. Lung emphysema is an abnormal air space enlargement, followed by destruction of the lung structure without any prominent fibrosis. From this perspective, one might say that in lung emphysema formation, any antifibrotic signal acts to prevent lung fibrosis. HLA-DOA is listed as a crucial gene related to molecular pathogenesis in activated pathways, such as the immune response IL-1 signaling pathway, positive regulation of smooth muscle migration, BMP signaling pathway, positive regulation of leukocyte migration, NF-kappB signaling, and cytochrome-c oxidase activity [46]

Line 462: FEN1 (Flap Structure-Specific Endonuclease 1), CDC20 (cell division cycle 20) and FERMT2 were downregulated by PFD, although only FERMT2 was upregulated in patients with IPF. Aging contributes to establishing IPF through altered gene expression related to mitochondrial function, cellular senescence, and telomeric length processes. FEN1, a gene associated with mitochondrial biogenesis and function, partially regulates the pathobiology of aging in IPF [47]. CDC20 acts as a regulatory protein interacting with several other proteins at multiple cell cycle stages. In a murine model of asbestos-induced fibrogenesis, several biological processes, including inflammation, proliferation, and matrix remodeling, were shown to be involved in developing and repairing asbestosis. CDC20 is a candidate gene for cell proliferation [48].

Comment 5

It is suggested that the authors explicitly address the comparison of the differentially expressed genes (DEGs) and key genes identified in the present study with the results reported in previous research analyzing gene expression in the context of IPF. Inclusion of this comparison in the Discussion section would be of vital importance, as it would allow contextualization and assessment of the consistency of the current findings within the broader landscape of IPF research, thus providing a more complete and grounded perspective on the relevant scientific literature.

Response 5

  The reviewer suggested that providing a more complete and grounded perspective on the relevant scientific literature would give us a contextualization and assessment of the consistency in this study. Our study is the first study, to our knowledge, to show the different mechanistic roles of the two antifibrotic drugs using PBMC transcriptome analysis. A previously published study entitled “Peripheral blood mononuclear cell gene expression profiles predict poor outcome in idiopathic pulmonary fibrosis” (Ref 1a) hypothesized that PBMC gene expression patterns may be predictive of poor outcomes in IPF patients, demonstrating that decreased expression of genes belonging to “The costimulatory signal during T cell activation” in particular CD28 was associated with shorter survival. One another study showed that monocytes from IPF patients displayed increased expression of CD64 (FcγR1) which correlated with amount of lung fibrosis, and an amplified type I IFN response ex vivo, indicating that a primed type I IFN pathway may contribute to driving chronic inflammation and fibrosis (Ref 2). These studies have expressed one aspect of IPF pathobiology, respectively, which are different from our research.

PBMCs are immune cells that are mainly classified into lymphocytes and monocytes. We hypothesized that PBMCs in IPF patients derive from bone marrow cells and the information carried is thought to be reflected by germline and somatic mutations related to IPF pathogenesis and specific treatment of anti-fibrotic drugs. These gene directives in PBMCs may play an important role in the further development of IPF pathology. It is plausible that the transcriptome of bulk PBMCs reflects certain aspects of the final lung reconstruction pathobiology. Regret to say that our study has only shown one side of IPF pathobiology.

Ref 1. Sci Transl Med. 2013 Oct 2;5(205):205ra136. doi: 10.1126/scitranslmed.3005964.

Peripheral blood mononuclear cell gene expression profiles predict poor outcome in idiopathic pulmonary fibrosis.

Ref 2. Front Immunol. 2021 Mar 5:12:623430.doi: 10.3389/fimmu.2021.623430.

Fraser E, et al. Multi-modal characterization of monocytes in idiopathic pulmonary fibrosis reveals a primed type I interferon immune phenotype.

Therefore, we have added the following paragraph in the discussion.

In the Discussion.

Line 269: To the best of our knowledge, the present study is the first to demonstrate the different mechanistic roles of these two antifibrotic drugs using PBMC transcriptome analysis. Herazo-Maya A et al. have hypothesized that PBMC gene expression patterns using microarray analyses may be predictive of poor outcomes in patients with IPF, demonstrating that decreased expression of genes belonging to “The costimulatory signal during T cell activation,” in particular CD28 was associated with shorter survival [12]. Fraser E et al. have reported that monocytes from patients with IPF displayed increased expression of CD64 (FcγR1), which correlated with the amount of lung fibrosis, and an amplified type I interferon response ex vivo, indicating that a primed type I interferon pathway may contribute to driving chronic inflammation and fibrosis [14].

Comment

Overall, I congratulate the authors for their commendable efforts in their search for cures for disease and improvement of human life. I wish them success in improving their research.

Response

 We appreciate the favorable and encouraging comments.

Reviewer 2 Report

Comments and Suggestions for Authors

Thank you for allowing me to review the paper. The authors examined the impact of anti-fibrotic drugs on the transcriptome of peripheral blood mononuclear cells in idiopathic pulmonary fibrosis (IPF). Gene expression levels were well-differentiated between IPF and healthy control, pre-treatment and post-treatment of PFD and NTD. The authors indicated the different role of the two antifibrotic drugs based on the gene expression profiles. Although this is an interesting finding, some major issues should be addressed to strengthen the critical aspects of this study.

Major

1. Figures 1-3 represent differentially expressed genes associated with IPF, Figures 4-6 represent genes associated with PFD, and Figures 7-9 represent genes associated with NTD. To assess the effects of PFD/NTD on peripheral blood mononuclear cells in IPF, a comparison should be made among healthy individuals, IPF patients (pre-treatment), and IPF patients (post-treatment). The authors need to focus on the gene expression clusters upregulated in IPF and downregulated after PFD or NTD treatment.

2. It would be of interest to see the pre- and post-treatment changes in known PFD target molecules (TGF-B1, TNF-a, IL-1B) and known NTD target molecules (VEGFR, PDGFR and FGFR) to understand their dynamics.

3. Although it might be challenging due to the small sample size, please address the relationship between the therapeutic effects and side effect profiles of PFD and NTD. It would be beneficial to present lung function (FVC, DLco) and chest HRCT findings for each case.

Minor

1. For volcano plot (Figure 2, 5, 8), please put some gene names on the plot for representative genes such as Serpine, Elovl6, Kank1, etc.

2. For heatmap analysis (Figure 3, 6, 9), please put the gene names on the right side.

Author Response

Response to the Comments of Reviewer 2

The reviewer raised several important points, thus the comments have been incorporated in the revised manuscript. We appreciate the thoughts and the time that the reviewer has given to this study. We hope the revisions and responses are acceptable for publication in IJMS.

The authors examined the impact of anti-fibrotic drugs on the transcriptome of peripheral blood mononuclear cells in idiopathic pulmonary fibrosis (IPF). Gene expression levels were well-differentiated between IPF and healthy control, pre-treatment and post-treatment of PFD and NTD. The authors indicated the different role of the two antifibrotic drugs based on the gene expression profiles. Although this is an interesting finding, some major issues should be addressed to strengthen the critical aspects of this study.

Comment 1

  Figures 1-3 represent differentially expressed genes associated with IPF, Figures 4-6 represent genes associated with PFD, and Figures 7-9 represent genes associated with NTD. To assess the effects of PFD/NTD on peripheral blood mononuclear cells in IPF, a comparison should be made among healthy individuals, IPF patients (pre-treatment), and IPF patients (post-treatment). The authors need to focus on the gene expression clusters upregulated in IPF and downregulated after PFD or NTD treatment.

Response 1

We appreciated the reviewer’s constructive comments regarding the methods of comparisons in the present study. We aimed to reveal different mechanistic roles for the two antifibrotic drugs with respect to PBMC transcriptome analysis. Since IPF covers a wide range of pathobiology, we were unable to present all about pathobiology. Therefore, to show the different mechanistic roles for the two antifibrotic drugs, we first compared PBMC transcriptome in IPF patients (pre-treatment of PFD or NTD) with those in HCs. Then, we compared PBMC transcriptome before and after anti-fibrotic drug treatment.

  The reviewer suggested that we should focus on the gene expression clusters upregulated in IPF and downregulated after PFD or NTD treatment. That is what we thought at first. However, during the analytic process, we have detected Gene Ontology terms (Biological Process) with upregulated genes such as Negative regulation of plasminogen activation (GO:0010757) and Negative regulation of cell adhesion mediated by integrin (GO:0033629) after PFD treatment. We thought “negative regulation” of upregulation are equivalent to downregulation. Therefore, we have included both GO terms of upregulation and downregulation after PFD or NTD treatment.

Comment 2

It would be of interest to see the pre- and post-treatment changes in known PFD target molecules (TGF-B1, TNF-a, IL-1B) and known NTD target molecules (VEGFR, PDGFR and FGFR) to understand their dynamics.

Response 2

In response to the reviewer’s comment, we have added the results in the manuscript and supplementary tables (Tables S6 and S11) regarding transcriptome levels of those known target molecules before and after PFD or NTD administration.

Table S6: Differences in transcriptome levels of molecules related to pirfenidone (PFD) acting mechanisms.

Table S11: Differences in transcriptome levels of molecules related to nintedanib (NTD) acting mechanisms.

Line 164: Transcriptome levels of TGF-β1, TNF-α, and IL-1B, the known molecules related to potential PFD acting mechanisms, were evaluated, and no significant differences were observed between those before and after PFD administration (Supplementary Table S6).

Line 217: The transcriptome levels of VEGFRs, PDGFRs, and FGFRs, known molecules related to potential NTD acting mechanisms, were evaluated, and no significant differences were observed between those of before and after NTD administration (Supplementary Table S11).

Comment 3

Although it might be challenging due to the small sample size, please address the relationship between the therapeutic effects and side effect profiles of PFD and NTD. It would be beneficial to present lung function (FVC, DLco) and chest HRCT findings for each case.

Response 3

We appreciated the reviewer’s constructive comments regarding the natural progression of IPF during the anti-fibrotic treatment. We have considered that IPF itself did not clinically progress as we have shown FVC data before and after anti-fibrotic drugs treatment, added to the Table 1. In addition, we have added the description regarding HRCT findings and side effects in the manuscript as described below. Unfortunately, DLco data before and after treatments was unavailable due to as an infection control during the COVID-19 pandemic.

In the results.

Line 82: No side effects were observed in patient sample No. 1, anorexia was detected in the patient of sample No. 2, and photosensitivity in the patient of sample No. 3. The NTD dose was 300 mg/day for patients of samples Nos. 4, 5, and 6. Side effects were diarrhea in patients of samples Nos. 1 and 2 and none in the patient of sample No. 3. Regarding the high-resolution computed tomography (CT) findings in terms of categorizing CT imaging in the evaluation of IPF, the usual interstitial pneumonia (UIP) pattern was observed in the patients of samples Nos. 1, 3, and 4, probable UIP pattern was observed in those of Nos. 2 and 5, and indeterminate for UIP was observed in the patient of sample No. 6. No apparent changes in forced vital capacity (FVC) or high-resolution CT findings were observed three months after the administration of either antifibrotic drug.

Comment 4

 For volcano plot (Figure 2, 5, 8), please put some gene names on the plot for representative genes such as Serpine, Elovl6, Kank1, etc.

Response 4

We again appreciated the reviewer’s constructive comments. We have put gene names on the plot regarding representative genes including Serpine and Elovl6 in Figures 2, 5, and 8 according to the reviewer’s suggestion.

Comment 5

For heatmap analysis (Figure 3, 6, 9), please put the gene names on the right side.

Response 5

We again appreciated the reviewer’s constructive comments. We have additionally prepared supplementary figures (Figures S1, S2, and S3) in which gene names on the heat maps have been presented according to the reviewer’s suggestion.

Reviewer 3 Report

Comments and Suggestions for Authors

In this manuscript, Ishii et al. analyze PBMC transcriptomes in healthy vs. IPF patients, and in patients before and after nintedanib or pirfenidone treatment.

The manuscript is well-written and addresses an important topic (i.e. IPF biomarkers), but the data are limited to transcriptomic analysis with no follow-up, mechanistic work, or validation, limiting the value of the work. The data and analysis represent a good start to this work but, in my opinion, this work is not yet worthy of publication without follow-up.

Specific major and minor comments:

Line 41, which FGFRs?

Authors should give more detail, especially in the results section, regarding the administration of pirfenidone and nintedanib. How long have these patients been on these regimens by the time their PBMCs are draw for RNA-seq analyses? What doses of these drugs are these patients on?

Authors describe DEGs when comparing PBMC transcriptomes before and after nintedanib or pirfenidone treatment, but IPF is a disease that progresses fairly quickly, so how do authors propose to disentangle which differential gene expression paradigms are simply a characteristic of worsening IPF, rather than of (presumably beneficial) drug effects?

In table 2, authors demonstrate pathway enrichment in DEGs, but only list one or two DEGs for each term. Are these the only dysregulated genes in these pathways? If so, especially, many of these terms are unlikely to be meaningful. E.g. “regulation of wound healing, spreading of epidermal cells.” has no relevance to PBMC phenotypes in IPF.

The authors spend much time in the discussion talking about the relevance of these terms and genes to IPF, potentially, but they are making dramatic assumptions, including that the DEG profiles result in clinically meaningful expression of certain genes (e.g. those encoding cytokines), that these differential transcript expressions result in differential protein expression and release (which would be better served through, for example, serum proteomics or connections to overexpression of said proteins in IPF tissue), and that overexpression of these genes in PBMCs would be relevant to pulmonary fibrosis in the same way that they might be relevant if they were found overexpressed in, for example, pulmonary fibroblasts (e.g. COL1A1, SERPINE1, etc.). To their credit, the authors somewhat address these assumptions in the discussion in lines 376-381, but this is such a strong limitation that I cannot conclude that their current data are meaningful without further exploration of these assumptions.

Author Response

Response to the Comments of Reviewer 3

General Comment

In this manuscript, Ishii et al. analyze PBMC transcriptomes in healthy vs. IPF patients, and in patients before and after nintedanib or pirfenidone treatment. The manuscript is well-written and addresses an important topic (i.e. IPF biomarkers), but the data are limited to transcriptomic analysis with no follow-up, mechanistic work, or validation, limiting the value of the work. The data and analysis represent a good start to this work but, in my opinion, this work is not yet worthy of publication without follow-up.

Response 

The reviewer raised several important points, thus the comments have been incorporated in the revised manuscript. We appreciate the thoughts and the time that the reviewer has given to this study. We hope the revisions and responses are acceptable for publication in IJMS.

Comment 1

Line 41, which FGFRs?

Response 1

It is FGFR 1,2, and 3. We have added the subtypes of FGFR in Line 41 as below.

In the revised manuscript

Line 41: fibroblast growth factor (FGF) receptors 1, 2, and 3.

Comment 2

Authors should give more detail, especially in the results section, regarding the administration of pirfenidone and nintedanib. How long have these patients been on these regimens by the time their PBMCs are draw for RNA-seq analyses? What doses of these drugs are these patients on?

Response 2

The reviewer asked the administration period and doses of pirfenidone and nintedanib in this study. PBMCs were collected about three months after the patient started taking antifibrotic drugs as described in Line 506. We have added this information regarding the drug doses as below.

Line 80: The dose of PFD was 1800 mg/day for the patient of sample No. 1 and 1200 mg/day for the patients of samples Nos. 2 and 3. No side effects were observed in patient sample No. 1, anorexia was detected in the patient of sample No. 2, and photosensitivity in the patient of sample No. 3. The NTD dose was 300 mg/day for patients of samples Nos. 4, 5, and 6.

Comment 3

  Authors describe DEGs when comparing PBMC transcriptomes before and after nintedanib or pirfenidone treatment, but IPF is a disease that progresses fairly quickly, so how do authors propose to disentangle which differential gene expression paradigms are simply a characteristic of worsening IPF, rather than of (presumably beneficial) drug effects?

Response 3

We appreciated the reviewer’s constructive comments regarding the natural progression of IPF during the anti-fibrotic treatments. IPF is a progressive fibrosing disease and it is not certain whether antifibrotic drugs exert a useful effect to suppress fibrosis as the reviewer pointed out. Therefore, we have added the changes of FVC (L) data before and after the administration of antifibrotic drugs to show that they may be effective in Table 1. No significant changes in FVC (L) were observed before and after antifibrotic treatment (Table 1), indicating an effect of antifibrotic drugs to suppress IPF progression. We suppose that the results of our transcriptome study would be reflected by one side of antifibrotic drug effects, since we could not deny that the results are somewhat reflected by natural progression of IPF. We have added the following sentence in the results section.

Line 89: No apparent changes in forced vital capacity (FVC) or high-resolution CT findings were observed three months after the administration of either antifibrotic drug.

Comment 4

In table 2, authors demonstrate pathway enrichment in DEGs, but only list one or two DEGs for each term. Are these the only dysregulated genes in these pathways? If so, especially, many of these terms are unlikely to be meaningful. E.g. “regulation of wound healing, spreading of epidermal cells.” has no relevance to PBMC phenotypes in IPF.

Response 4

  We appreciated the reviewer’s constructive comments regarding the pathway enrichment and DEGs.

The gene ontology terms, presented in table 2, are not necessarily associated with the pathobiology of idiopathic pulmonary fibrosis. For example, as the reviewer indicated, “regulation of wound healing, spreading of epidermal cells” has no relevance to PBMC phenotypes in IPF. The number of DEGs among GO term or KEGG pathway are few, therefore we have focused on the DEGs as genes that were detected in the setting of fold change setting of > 2 or < 0.5, and p-value < 0.05. If the fold change and statistically significant p-value in the analysis setting were set less severe, we expect more DEGs to be detected and those pathways would be more enriched. In the discussion part, we have focused on DEGs which would be potentially related to IPF pathobiology.

Comment 5

  The authors spend much time in the discussion talking about the relevance of these terms and genes to IPF, potentially, but they are making dramatic assumptions, including that the DEG profiles result in clinically meaningful expression of certain genes (e.g. those encoding cytokines), that these differential transcript expressions result in differential protein expression and release (which would be better served through, for example, serum proteomics or connections to overexpression of said proteins in IPF tissue), and that overexpression of these genes in PBMCs would be relevant to pulmonary fibrosis in the same way that they might be relevant if they were found overexpressed in, for example, pulmonary fibroblasts (e.g. COL1A1, SERPINE1, etc.). To their credit, the authors somewhat address these assumptions in the discussion in lines 376-381, but this is such a strong limitation that I cannot conclude that their current data are meaningful without further exploration of these assumptions.

Response 5

We have totally agreed with the reviewer’s deep insight, since we have had exactly the same opinion, considering that our discussion have been configured on big assumptions according to the reviewer’s indication. Therefore, we have added the limitation of this study in the discussion section.

Line 471: This study had several limitations. First, we considered possible mechanistic differences in the functions of these two antifibrotic drugs based on step-by-step assumptions. It was assumed that the DEG profiles result in the clinically meaningful expression of certain genes, including those encoding cytokines. Moreover, this differential transcript expression may result in differential protein expression and release. Furthermore, we assumed that overexpression of these genes in PBMCs would be relevant to pulmonary fibrosis in the same way that they might be relevant if overexpressed in pulmonary fibroblasts (e.g. COL1A1 and SERPINE1). Therefore, we believe that the results of the present study, which demonstrated that three months of antifibrotic treatment with PFD or NTD affected RNA-seq analysis of peripheral blood mononuclear cells, would have some pathobiological implications and suggest possible mechanistic differences in the functions of these two antifibrotic drugs. However, we could not prove the validity of these results in a clinical setting.

Round 2

Reviewer 1 Report

Comments and Suggestions for Authors

I sincerely appreciate the attention given to my comments and suggestions regarding the manuscript entitled "Effects of Antifibrotic Drugs on Transcriptome of Peripheral Blood Mononuclear Cells in Idiopathic Pulmonary Fibrosis". I have carefully reviewed the responses provided by the authors to my observations and comments. Below are my comments on the authors' responses and how they addressed each of the points I made:

Overall, I appreciate the responsiveness of the authors to my comments and observations. I believe they have adequately addressed each of my points, improving the quality and clarity of the manuscript. I believe that these modifications strengthen the validity and potential impact of the research presented. I recommend acceptance of the manuscript for publication.